# The Microstructural Evolution and Grain Growth Kinetics of Fine-Grained Extruded Mg-Nd-Zn-Zr Alloy

**DOI:** 10.3390/ma15103556

**Published:** 2022-05-16

**Authors:** Xueyan Jiao, Xinjie Li, Liqiang Zhan, Gang Wang, Jin Ding, Jianlei Yang

**Affiliations:** 1School of Naval Architecture and Port Engineering, Shandong Jiaotong University, 1508 Hexing Road, Weihai 264310, China; lxj17864736566@163.com (X.L.); hit105@163.com (J.D.); 2Weihai Lightweight Materials and Forming Engineering Research Center, Weihai 264209, China; happyflower1221@163.com (L.Z.); wg@hitwh.edu.cn (G.W.); jlyang@hit.edu.cn (J.Y.)

**Keywords:** Mg-Nd-Zn-Zr alloy, heat treatment, microstructure evolution, grain growth kinetics, abnormal grain growth

## Abstract

The microstructure evolution and grain growth kinetics of the fine-grained extruded Mg-Nd-Zn-Zr alloy were investigated by holding the extruded plate for a wide range of time in the temperature range of 470 °C to 530 °C. By observing the optical micrographs, it was found that the material showed abnormal grain growth at the experimental condition of 470 °C × 24 h, and the time point of abnormal grain growth appeared significantly earlier with the increase in the experimental temperature. The evaluation of the second phase content within the alloy indicates that the presence of the second phase contributes to the microstructural stability of the Mg-Nd-Zn-Zr alloy. However, the slow coarsening/dissolution of the second phase is an important cause of abnormal grain growth. Based on the experimental data, the isothermal grain growth kinetic models of the fine-grained extruded Mg-Nd-Zn-Zr alloy were developed based on the Sellars model. The grain growth exponent was in the range of 5.5–8 and decreased gradually with the increase in the experimental temperature. The grain growth activation energy is approximately 150.00 kJ/mol, which is close to the bulk diffusion activation energy of magnesium, indicating that the grain growth is controlled by lattice diffusion. By energy spectrometry (EDS), the compositional changes of the second phase within this alloy at 500 °C were investigated.

## 1. Introduction

Although traditional magnesium alloys have the characteristics of low density, high specific strength, and specific stiffness, their application in aerospace and other fields is limited due to their poor high-temperature stability [1,2]. The related studies have indicated that the addition of rare-earth elements (such as Ce, Nd, Gd, and La) to magnesium alloys not only significantly improves the high-temperature properties and refines the dendritic structure of the as-cast alloy [3,4] but also weakens the basal texture and the corresponding anisotropy [5,6].

Mg-Nd-Zn-Zr series alloys are the most extensively used group of rare earth magnesium alloys in the manufacturing industry. Basically, as-cast Mg-Nd-Zn-Zr alloy is mainly composed of α-Mg matrix and Mg_12_Nd eutectic compounds along with the grain boundaries [7,8]. A large number of second phases exist at the grain boundary, which can cause local stress concentration. These eutectic compounds are easy to form cracks, resulting in poor plasticity of as-cast alloy [9]. However, thermo-plastic processing (for example hot extrusion [10], hot rolling [11], etc.) not only breaks up the coarse internal eutectic compounds within the as-cast Mg-Nd-Zn-Zr alloy but also leads to grain refinement, which further improves the alloy strength and toughness. For example, Wang et al. [12] found that the grain size of Mg-2.5Nd-0.5Zn-0.5Zr can be refined to 4.25 mm after multi-pass hot rolling. The subsequent heat treatment is very important to improve the properties of deformed materials. However, the refinement of the grains will increase the free energy of the system, causing the alloy to be prone to grain growth at high temperatures [13]. There are two grain growth modes that have been accepted: (I) normal grain growth (NGG) and (II) abnormal grain growth (AGG) [14]. This will cause a significant decrease in strength and plasticity during the subsequent heat treatment or thermo-mechanical processing. On the other hand, the broken or precipitated second phases have good thermal stability, which allows the alloy to maintain good microstructure stability during subsequent heat treatment or thermo-mechanical processing [15]. Therefore, the study of grain growth behavior, under a high temperature environment and its key influencing factors, are important to achieve prediction and control of microstructural evolution.

The purpose of this work is to explore the microstructure evolution of fine-grained extruded Mg-Nd-Zn-Zr alloys and their grain growth kinetics when held at high temperatures for long periods of time.

## 2. Experimental

### 2.1. Alloy Preparation

The material used in this work was the extruded Mg-Nd-Zn-Zr alloy sheets with a cross-sectional size of 80^W^ mm × 5^t^ mm. The chemical composition is listed in Table 1 [11]. The extrusion temperature, the extrusion ratio, and die-exit speed were 400 °C, 19.63, and 1.5 mm/s, respectively. Subsequently, the extruded samples (size 10^L^ mm × 8^W^ mm × 5^t^ mm) were cut using wire cutting machine and then heat treated at 470 °C, 500 °C, and 530 °C for 10 min/40 min/1 h/2 h/4 h/6 h/8 h/12 h/24 h. It is worth noting that all samples were cooled to room temperature in air after finishing the heat treatment.

### 2.2. Microstructure Analysis

Microstructural examination was performed using X-ray diffraction (XRD, Bruker D8 Discover powder, Karlsruhe, Germany), an Optical Microscope (OM, DSX510, Tokyo, Japan), and Scanning Electron Microscopy (SEM, Zeiss MERLIN Compact, Oberkochen, Germany) equipped with an energy dispersive spectroscopy (EDS) detector. The samples were ground and polished, and then, they were etched using an etching agent (picric acid (6 g) + acetic acid (5 mL) + water (10 mL) + alcohol (100 mL)) for 8–10 s. The grain size was counted according to Nano Measurer 1.2, and five metallographic organization photos were counted for each state, with about 300 grains counted per photo. The second phase content in each state was counted by Image-Pro Plus 6.0 software(Media Cybernetics, Washington, DC, USA), and five photos were counted in each state using three samples. The dimensions and histograms of the grain size were measured using the interception method.

### 2.3. Grain Growth Kinetics

Grain growth is a thermally activated process related to atomic diffusion and interfacial reactions of the material [15,16]. To evaluate the grain growth kinetics and achieve a prediction of the grain growth behavior at high temperatures, the Sellars equation was fitted to the grain growth process. According to the Sellars equation, the average grain size of the alloy, at constant temperature versus holding time, can be expressed as [17]:(1)$$D^{n}-D_{0}^{n}=kt$$
where *D* is the measured average grain size, *D*_0_ is the average initial grain size, *n* is the grain growth exponent, the constant k is termed the growth rate related to grain boundary mobility, and *t* is the holding time. The constant k is defined as:(2)$$k=k_{0}\exp\left(-\frac{Q}{RT}\right)$$
where k_0_ is the pre-exponential constant, *Q* is the grain growth activation energy, *R* is the gas constant, and *T* is the absolute temperature of heat treatment. By differentiating Equation (1), the grain growth rate i.e., *dD*/*dt* was obtained. The grain growth rate can be written as:(3)$$\frac{dD}{dt}=\frac{k}{nD^{n-1}}$$

To evaluate the value of *dD*/*dt* accurately, the measured average grain size was fitted to the holding time in segments by a nonlinear regression method. The obtained curves were then differentiated to obtain the growth rate *dD*/*dt* for each heat treatment state [18]. Subsequently, Equation (3) is taken in logarithmic form
(4)$$\ln\left({\frac{dD}{dt}|}_{T=const}\right)=-\left(n-1\right)\ln\left(D\right)+\ln\left(\frac{k}{n}\right)$$

## 3. Results and Discussion

### 3.1. The Initial Microstructure

The microstructure of the Mg-Nd-Zn-Zr alloy, before and after extrusion, is shown in Figure 1. After preheating at 400 °C for 9 h, a large number of Mg_12_Nd phases were still preserved at the grain boundaries, and the rod-like Zr-Zn phases were produced inside the grains [7,19,20] (Figure 1a). In contrast, after extrusion at 400 °C, the internal microstructure was more homogeneous, and the grains were significantly refined (Figure 1b). Using the linear intercept method, the average grain size was about 2.78 ± 0.31 μm. The corresponding histogram of the grain size distribution is shown in the upper right corner of the micrograph. In addition, a large number of second phases were generated at the grain boundaries with the grain interior, and most of them are Mg_12_Nd phases (Figure 1c). Their origin can be roughly divided into two parts: Part 1, which is produced by the second phase that was not dissolved in the matrix inside the alloy before extrusion and was broken during extrusion; Part 2, which precipitated from inside the matrix during extrusion. Regardless of which way the second phase is obtained, it has a strong pegging effect on the grain boundaries and can improve the organizational stability of the alloy at high temperatures [16].

### 3.2. Dependence of Microstructure on Heat Treatment Condition

Figure 2 illustrates the microstructural evolution during heat treatment at 470 °C for holding times of 10 min, 40 min, 1 h, 2 h, 4 h, 6 h, 8 h, 12 h, and 24 h. The corresponding histograms of the grain size distribution are also shown in the upper right corner of the micrograph. It is noticeable that the mean grain size gradually increases with an increased holding time. Meanwhile, the grains were more uniform under each holding time, but a small number of coarse grains appeared after 24 h of holding, indicating that normal grain growth (NGG) was dominant during the heat treatment at 470 °C, and the phenomenon of abnormal grain growth appeared in 24 h of holding.

Figure 3 shows the evolution of grain growth at heat treatment temperature of 500 °C for holding times of 10 min, 40 min, 1 h, 2 h, 4 h, 6 h, 8 h, 12 h, and 24 h. With extending soaking time, the average grain size grows up from the premier value of 2.78 ± 0.31 μm (Figure 1b) to about 9.09 ± 0.12 μm for 10 min, 15.60 ± 0.06 μm for 8 h, and 17.51 ± 0.17 μm for 24 h. Furthermore, after 12–24 h of holding time, large-sized grains were found and their number gradually increased, which indicates that abnormal grain growth (AGG) began to occur after 12 h of holding time. This is due to the second phase in the grain boundary being redissolved, and the migration resistance of grain boundary could be decreased and led to some abnormal grain growth.

The optical micrographs of the samples’ heat treatment at 530 °C for 10 min, 40 min, 1 h, 2 h, 4 h, 6 h, 8 h, 12 h, and 24 h, with associated grain size distribution histograms, are shown in Figure 4. Consistent with 470 °C and 500 °C, the average grain size increased with the extension of holding time when heat treatment was performed at 530 °C. It can be found, from the optical micrographs, that the grain size is more uniform when held at 530 °C for 10 min–2 h, while the uniformity of the grains decreases with further extension of the holding time. The above phenomenon indicates that the transformation of NGG to AGG occurred during the holding time of 2–4 h.

In order to analyze the growth trend of average grain size at different heat treatment temperatures more visually, the curves of average grain size versus holding time pieces were drawn based on the above data, as shown in Figure 5. As seen, when heat treatment was performed at 470 °C and 500 °C (Figure 5a,b), the trend of grain growth was basically the same, and both had a rapid increase in the average grain size within a holding time of 10 min–2 h. This is due to the fact that, after extrusion, the internal grains of the alloy are significantly refined and the interfacial energy is significantly increased, while the reduction in the system interfacial energy is the main driving force for the grain growth [14], which results in the rapid increase in the internal grain size within 10 min–2 h. It is also due to the above reason that the average grain size inside the alloy has grown to 14.69 μm after only 10 min of holding at 530 °C (Figure 5c). With the extension of the holding time, the average grain size of the alloy grows slowly in 2–6 h. The reason is that the second phase inside the alloy dissolves into the matrix slowly at 470 °C and 500 °C, and the second phase that fails to dissolve into the matrix will hinder the growth of the grains, thus causing the slow increase in the grains in this time period. In general, the higher the temperature, the higher the grain growth rate. On the other hand, for a given temperature, the grain growth rate decreases with increasing annealing time due to the decrease in the interfacial energy of the system.

### 3.3. Dependence of Phase Content on Heat Treatment Regime

As mentioned in the above analysis, the presence of the second phase will hinder the grain growth. For this reason, the BSE photographs of different heat treatment states were obtained using SEM. After that, the second phase, and its volume fraction in each state, were observed and counted in combination with the image processing software, and the results are shown in Figure 6 and Figure 7. As seen, the volume fraction of the second phase inside the alloy gradually decreases with the increase in temperature and the extension of holding time. Moreover, by comparing the BSE photographs for a given soaking time, it is clear that the second phase is more sensitive to temperature. For example, the volume fraction of the second phase decreased to 6.32%, 2.97%, and 0.1% after holding at 470 °C, 500 °C, and 530 °C for 8 h, respectively. From Figure 7a,b, and combined with Figure 5a,b, it is interesting to note that the volume fraction of the second phase decreases slowly, within 2–6 h of holding, at 470 °C and 500 °C. Meanwhile, the growth rate of the grains is equally slow. This further proves that the fine and diffuse second phase has a pegging effect on the grain boundaries, which is beneficial to improve the microstructure stability of the alloy at high temperatures.

According to Equation (4), when the heating temperature is constant, ln(d*D*/d*t*) is linearly related to ln(*D*), so the value of *n* and k can be easily obtained from the slope of ln(d*D*/d*t*) vs. ln(*D*) using the least squares linear regression method (Figure 8). Based on the obtained *n* and k values, the average grain size under different heat treatment conditions was calculated theoretically, and the results are shown in the red curve in Figure 5. As can be seen from Figure 5, the average grain size, calculated theoretically, is closest to the actual measured value only at 530 °C, and the grain growth trend is basically the same. This is due to the rapid dissolution of the second phase into the matrix at 530 °C and the rapid weakening of the pegging effect on the grain boundaries, resulting in a grain growth trend close to the theoretical model.

To obtain the activation energy for grain growth of the extruded Mg-Nd-Zn-Zr alloy, Equation (2) is taken in a logarithmic form:(5)$$R\ln k=-\frac{Q}{T}+R\ln k_{0}$$

According to Equation (5), the curve of *R*ln(k) vs. l/*T* is then obtained (Figure 9). The value of *Q* and k_0_ can be determined by the slope and intercept in the plot of *R*lnk vs. 1/T, as shown in Figure 9. However, since only three experimental temperatures were set in this study, only a rough estimate of *Q*, *Q* ≈ 150.00 kJ/mol, could be made.

### 3.4. Phase Transition

As mentioned above, in addition to the grain growth within the alloy at high temperatures, there is also a process of dissolution of the second phase into the matrix. However, both the grain growth and the dissolution of the second phase into the matrix need to be accomplished by atomic diffusion. This inevitably results in the transformation of the second phase. Therefore, it is necessary to investigate the transformation of the second phase in Mg-Nd-Zn-Zr alloys under high temperature conditions.

To characterize the diffusion behavior of the elements during the heat treatment, SEM-EDS was used. Figure 10 shows the SEM-EDS point and map-scan results of the initial extruded state, as well as after holding at 500 °C for 8 h, 12 h, and 24 h, respectively. Table 2 shows the composition of the intermetallic phases obtained from SEM-EDS analysis, as illustrated in Figure 10. Among them, areas A, E, K, and R are the matrices, and the atomic content of Mg elements, in all the above regions, is higher than 99.50%. As seen in Figure 10, the Mg alloy in the initial extruded state contains two morphologies of the second phase: a massive second phase of large size (areas B and D); a spherical second phase of small size (area C). The large-sized second phase contains three elements: Mg, Nd, and Zn. The small-sized second phase contains two elements: Mg and Nd, while the Zr elements are uniformly distributed inside the alloy. Subsequently, after holding at 500 °C for 8 h, the number of small-sized second phases decreased significantly, while adjacent large-sized second phases merged with each other and led to an increase in the elemental Nd and Zn content (areas F, G, and J). In addition, a new Zr-Zn phase (area H) was found inside the alloy, which has a rod-like shape [7,18]. Continuing to extend the holding time, the small-sized second phase inside the alloy basically disappears, while the size of the large-sized second phase and the content of Nd and Zn elements gradually decreases. Meanwhile, three elements—Mg, Nd, and Zr—were found in the areas M and P. Since the electro-negativities of Mg and Zr are very close, and Nd elements are mostly combined with Mg elements, the Zr elements at these areas should come from the α-Zr phase.

## 4. Discussion

### 4.1. The Abnormal Grain Growth

According to Figure 1b, it can be seen that the Mg-Nd-Zn-Zr alloy has fine and uniform grains after extrusion, and no coarse grains exist. During the heat treatment process, with the increase in temperature and the extension of holding time, the grains grew gradually, and abnormally coarse grains appeared.

To distinguish AGG from NGG, Rios [21] proposed a criterion which states that AGG will result when the relative growth rate of the candidate abnormally grown grains, *d*(*D*_ad_/*D*_av_)/*dt*, is >0, where *D*_ad_ is the grain size of the candidate grain, and *D*_av_ is the measured average grain size. However, according to another criterion, AGG commences when the ratio of *D*_max_/*D*_av_ is greater than 5, where *D*_max_ is the maximum grain size measured [22,23,24]. Therefore, the maximum grain size (*D*_max_) and the ratio of *D*_max_/*D*_av_ for different heat treatment states were counted and calculated separately, and the results are shown in Table 3. As seen, the *D*_max_/*D*_av_ ratio generally lies between 2.5 and 5, while at the heat treatment states, identified as the beginning of AGG (in Section 3.1), the *D*_max_/*D*_av_ ratio is ≈4. The *D*_max_/*D*_av_ at the beginning of AGG is less than the value (*D*_max_/*D*_av_ > 5) specified in the standard. This may be due to the influence of a large number of second phases inside the alloy, which leads to the above phenomenon. On the one hand, the second phase in Mg alloys has a good thermal stability and can be pinned at grain boundaries to hinder grain growth [15,24]. On the other hand, the slow dissolution/roughing of part of the second phase is more likely to lead to the occurrence of AGG [24,25,26]. In addition, in combination with Figure 4, unusually coarse grains appeared at 530 °C for 4 h, and the ratio of *D*_max_/*D*_av_ was close to 4. However, as the holding time was extended to 6–8 h, the grains gradually became homogeneous, and the ratio of *D*_max_/*D*_av_ gradually decreased. Subsequently, the abnormally coarse grains reappeared after the extension of the holding time. This observation suggests that a temporary transition from AGG to NGG occurred [23,27]. The authors of [24] attributed this phenomenon to the completion of secondary recrystallization.

### 4.2. Analysis of Grain Growth Kinetics

Regarding the grain growth exponent *n*, Burke et al. [28] obtained a grain growth exponent of 2 under ideal conditions by assuming that the driving force of grain growth is only influenced by the boundary curvature. Obviously, in actual materials, the presence of second-phase particles causes the experimental predictions to deviate from the theoretical values. Related studies found that the grain growth exponent of magnesium alloys usually lies in the range of 2–8 [29,30,31]. As shown in Figure 8, the grain growth exponent of the extruded Mg-Nd-Zn-Zr alloy in this study lies roughly in the range of 5.5–8. Since the *n*-value is a measure of the resistance to grain boundary motion, such a high *n*-value indicates that the grain growth process is hindered to a large extent [30]. On the one hand, it is due to the pegging effect of the second phase particles at the grain boundaries, which leads to a greater resistance to grain boundary movement. On the other hand, it may be due to the strong initial texturing of the Mg-Nd-Zn-Zr alloy itself after extrusion, which leads to the possible enhanced effect of anisotropy on the boundary energy and mobility [11,29]. Furthermore, as seen in Figure 8, the *n*-value of the extruded Mg-Nd-Zn-Zr alloy gradually decreases with the increase in the experimental temperature, which, combined with Figure 6 and Figure 7, can be inferred to be due to the decrease in the second phase particles, which decreases the resistance during grain growth and, thus, leads to the decrease in the *n*-value.

The grain growth activation energy determined from Figure 9, *Q* ≈ 150.00 kJ/mol, is higher than that of bulk diffusion in Mg (135 kJ/mol) [32]. Considering the experimental conditions and the inherent errors in the theoretical calculations, this suggests that the grain growth activation energy of the extruded Mg-Nd-Zn-Zr alloy is close to the bulk diffusion activation energy. However, related studies have shown that the grain growth activation energy of Mg alloys with low alloying elements (e.g., pure Mg, AZ31) is closer to the grain boundary diffusion activation energy of Mg (92 kJ/mol) [20,29,32]. This has been attributed to the lower second phase content and the finer initial grain size. Conversely, the presence of a large number of second-phase particles in the extruded Mg-Nd-Zn-Zr alloy will lead to an increase in the grain growth activation energy. On the other hand, the atomic radius of Nd (*r* = 1.82) is higher than that of Mg (*r* = 1.60), and a certain lattice distortion will occur during the dissolution of Nd elements into the Mg matrix, which will hinder the lattice diffusion. Thein [33] et al. similarly found that the addition of Ti atoms increased the grain growth activation energy of the Mg-5 wt%Al alloy and close to the bulk diffusion activation energy. This indicates that bulk diffusion, rather than grain boundary diffusion, is the dominant mechanism in the temperature range studied.

### 4.3. Dissolution Sequence of the Second Phase

Related studies have shown that the addition of Nd elements can give Mg alloys a significant age strengthening effect [33,34]. To this end, the researchers obtained the phase precipitation sequence of Mg-Nd binary alloy during aging [34,35,36]: SSSS → G.P. Zones → *β*″(DO_19_) → *β*’(Mg_7_Nd) → *β*_1_(Mg_3_Nd) → *β*(Mg_12_Nd) → *β_e_*(Mg_41_Nd_5_). On the other hand, other researchers found that the addition of appropriate Zn elements to the Mg-Nd alloy could further improve the age-strengthening effect of the Mg alloy [37]. The addition of Zn elements also changed the phase precipitation sequence during aging [36,38]: SSSS → G.P. Zones → *γ*″(Mg_7_(Nd,Zn)_3_) → *γ*, where the sub-stable γ″ phase has a hexagonal structure (a = 0.556 nm, c = 1.563 nm) and precipitates as a platelet on the base of α-Mg, while the steady-state γ phase is a Mg-Nd-Zn ternary compound with a fcc structure (a = 0.7444 nm) [39]. Conversely, a similar phase transition is bound to occur when the phase is dissolved into the α-Mg matrix at high temperature. For this reason, the dissolution process of phases in Mg-Nd-Zn-Zr alloys, at high temperatures, is discussed in this study.

According to the data in Figure 1c and Figure 10 and Table 2, the initial extruded Mg-Nd-Zn-Zr alloy contains not only a large amount of fine Mg_12_Nd phases but also Mg-Nd-Zn ternary phases of larger size (regions B and D in Figure 10). Subsequently, after holding at 500 °C for 8 h, the fine Mg_12_Nd phase decreased significantly, while merging occurred between adjacent Mg-Nd-Zn (*γ*″ phase [40]) ternary phases. With the extension of the holding time, the Nd element in the Mg-Nd-Zn ternary phase decreased simultaneously with the Zn element, and after 12 h of holding, part of the Mg-Nd-Zn ternary phase was transformed into the Mg-Nd binary phase with a significant decrease in quantity. Therefore, the coarsening/dissolution of the second phase resulted in the abnormal grain growth (AGG) of the extruded Mg-Nd-Zn-Zr alloy after 12 h of holding at 500 °C. Continuing to extend the holding time to 24 h, a large number of needle-like Mg-Nd (Mg_5_Nd phase [41]) binary phases appeared inside the alloy. In summary, the following dissolution order of the second phase within the alloy may exist when the extruded Mg-Nd-Zn-Zr alloy is heat treated at 500 °C: Mg-Nd-Zn ternary phase (*γ*″ phase) → bulk Mg-Nd binary phase (to be determined) → needle Mg-Nd binary phase (Mg_5_Nd phase). However, the above dissolution sequence is a speculation based on the available data, and the exact phase dissolution sequence of Mg-Nd-Zn-Zr alloys at high temperatures still needs further investigation.

## 5. Conclusions

By studying the microstructure and grain growth kinetics of the fine-grained extruded Mg-Nd-Zn-Zr alloy, during heat treatment at 470 °C–530 °C, the following conclusions can be drawn.

(1) When the extruded Mg-Nd-Zn-Zr alloy with an initial average grain size of 2.78 μm was heat treated at 470 °C–530 °C, it was found that the average grain size gradually increased with increasing experimental temperature and holding time, while the second phase content gradually decreased. Due to the increase in temperature, the grain growth rate increased rapidly, but due to the decrease in interfacial energy, the grain growth rate decreased gradually with the increase in holding time.

(2) On the other hand, the initial extruded Mg-Nd-Zn-Zr alloy contains a large amount of the second phase inside, resulting in relatively slow grain growth at lower temperatures (470 °C–500 °C). Moreover, the coarsening/dissolution of the second phase leads to a change in the grain growth pattern, from NGG to AGG, observed at the heat treatment conditions of 470 °C × 24 h, 500 °C × 12 h, and 530 °C × 4 h. This was confirmed by calculating the *D*_max_/*D*_av_ for the different heat treatment states.

(3) The grain growth exponent of the extruded Mg-Nd-Zn-Zr alloy (*n* ≈ 5.5–8) is higher than the theoretical value (*n* = 2) due to the presence of a large amount of the second phase inside the alloy, and the dissolution of the second phase causes the *n*-value to decrease gradually with the increase in experimental temperature. Meanwhile, the grain growth activation energy is roughly estimated to be 150.00 kJ/mol, which is close to the bulk diffusion activation energy of magnesium.

(4) According to EDS observations, the initial extruded Mg-Nd-Zn-Zr alloy contains not only the Mg12Nd phase but also the *γ*″ phase, and the following phase dissolution sequence may occur when heat treatment is performed at 500 °C: Mg-Nd-Zn ternary phase (*γ*″ phase) → bulk Mg-Nd binary phase (to be determined) → needle Mg-Nd binary phase (Mg_5_Nd phase).

## Figures and Tables

**Figure 1 materials-15-03556-f001:**
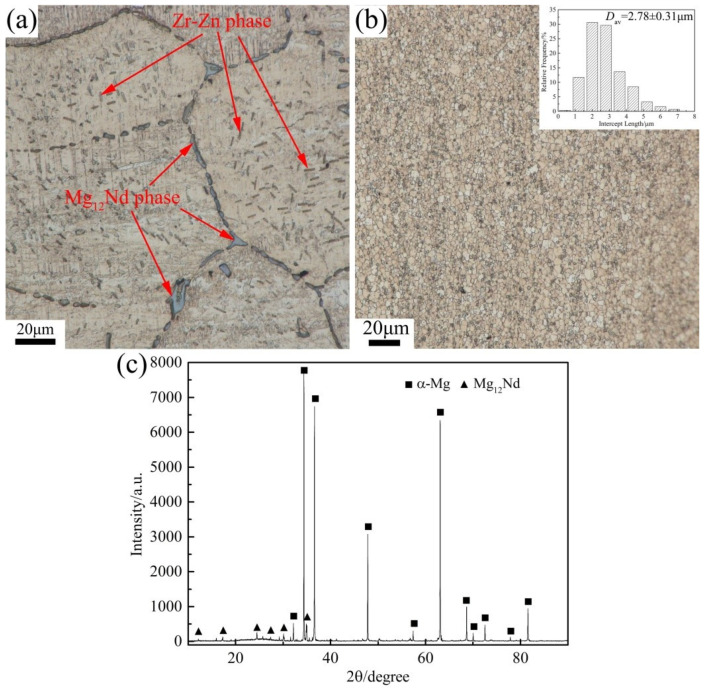
Optical micrograph of the alloy (**a**) as-cast after preheating at 400 °C for 9 h, (**b**) after extrusion at 400 °C, and (**c**) its corresponding XRD patterns.

**Figure 2 materials-15-03556-f002:**
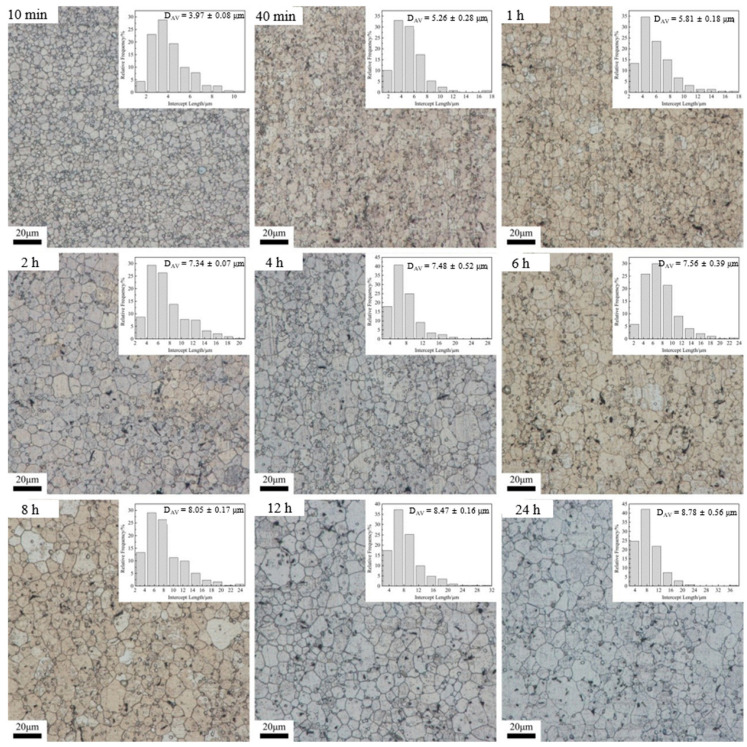
Optical micrographs and the corresponding grain-size distribution histograms obtained upon heat treatment at 470 °C.

**Figure 3 materials-15-03556-f003:**
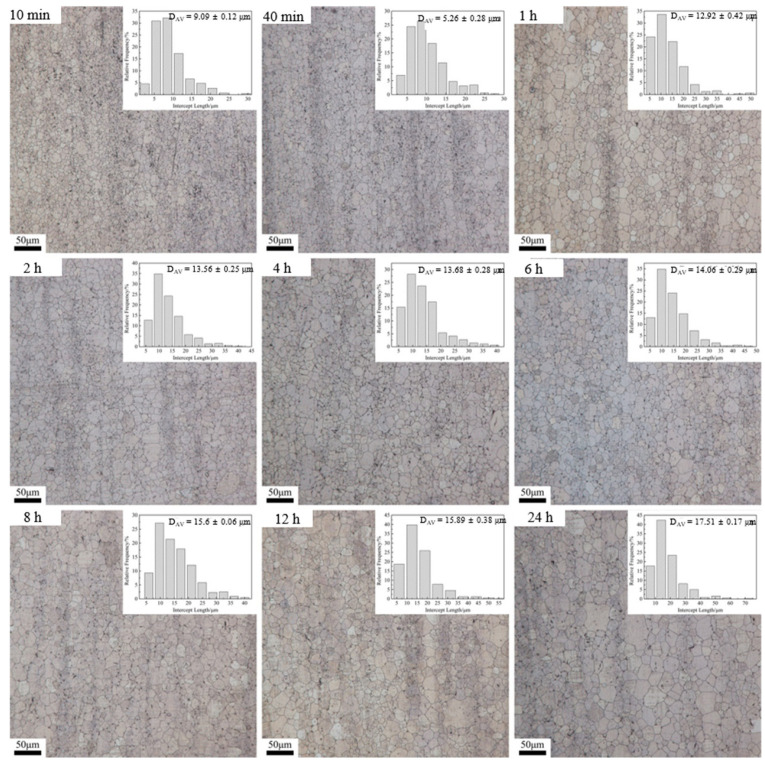
Optical micrographs and the corresponding grain-size distribution histograms obtained upon annealing at 500 °C.

**Figure 4 materials-15-03556-f004:**
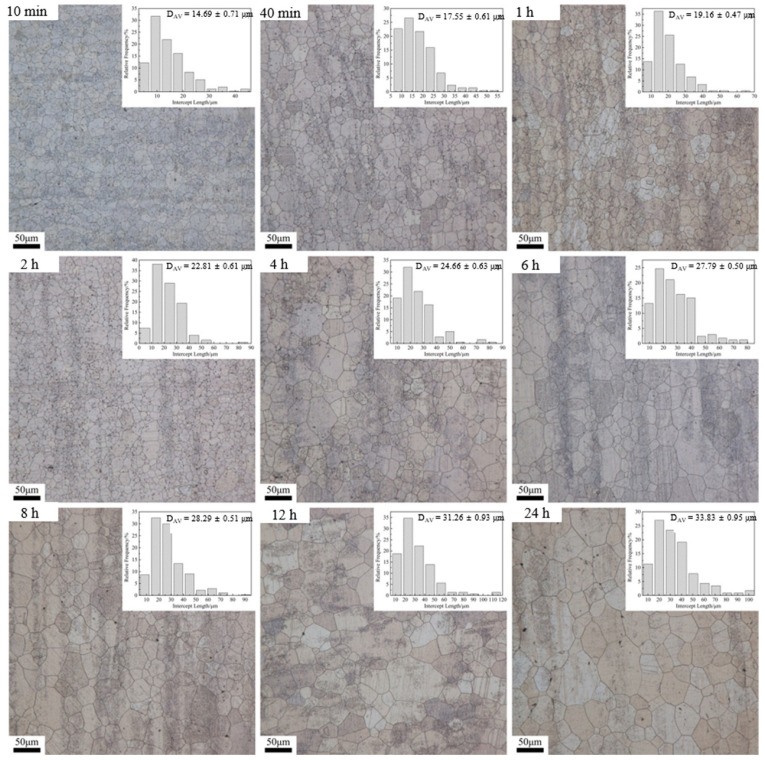
Optical micrographs and the corresponding grain-size distribution histograms obtained upon annealing at 530 °C.

**Figure 5 materials-15-03556-f005:**
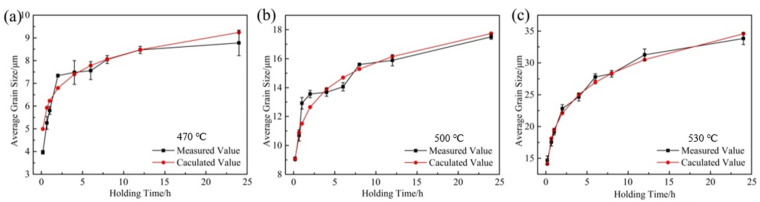
The curves of average grain size versus holding time pieces at (**a**) 470 °C, (**b**) 500 °C, and (**c**) 530 °C.

**Figure 6 materials-15-03556-f006:**
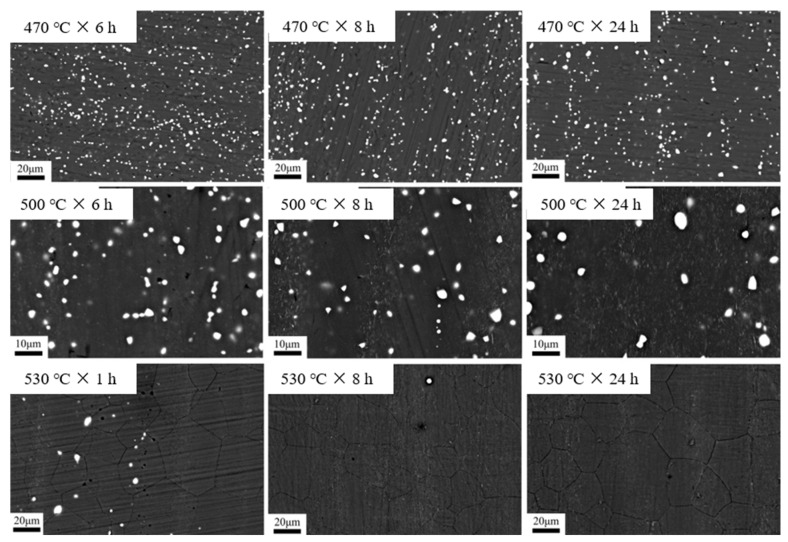
The BSE photographs of different heat treatment states.

**Figure 7 materials-15-03556-f007:**
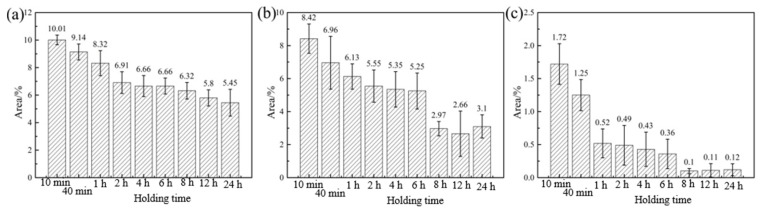
The volume fraction of the second phase against holding time at (**a**) 470 °C, (**b**) 500 °C, and (**c**) 530 °C.

**Figure 8 materials-15-03556-f008:**
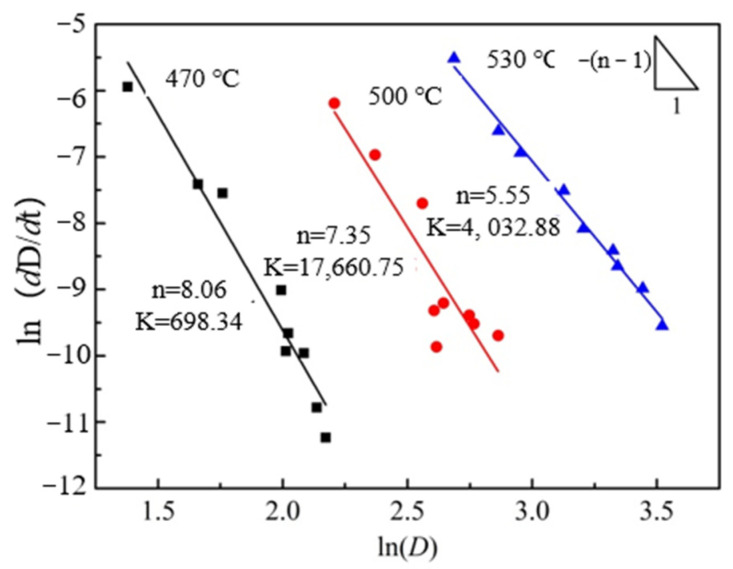
Relationship of the logarithm of (d*D*/d*t*) vs. the logarithm of *D*.

**Figure 9 materials-15-03556-f009:**
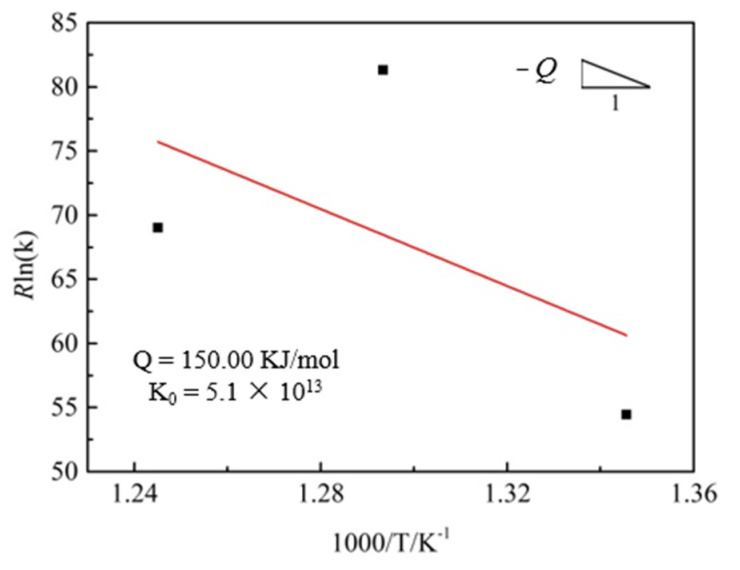
Relationship of *R*ln(k) vs. 1/*T*.

**Figure 10 materials-15-03556-f010:**
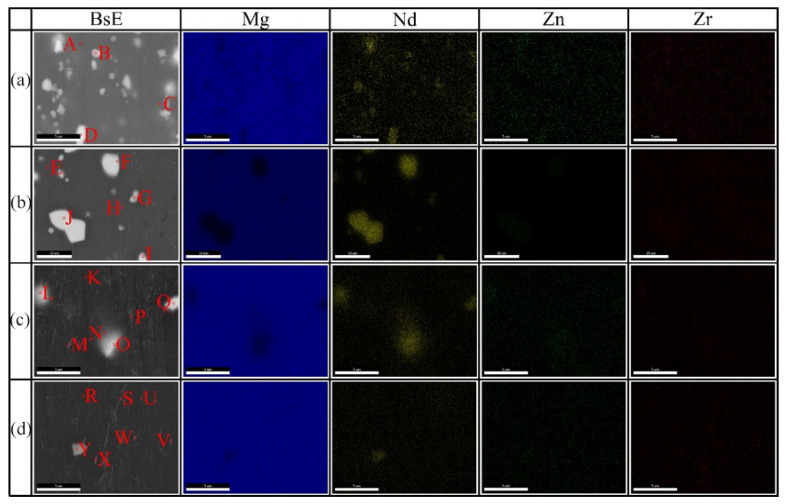
The SEM-EDS element mapping of the initial extruded state and after holding at 500 °C for 8 h, 12 h, and 24 h, respectively. (**a**) As-extruded, (**b**) 500 °C × 8 h, (**c**) 500 °C × 12 h, (**d**) 500 °C × 24 h.

**Table 1 materials-15-03556-t001:** The chemical composition of as-extruded Mg-Nd-Zn-Zr alloy (wt%) [11].

Element	Nd	Zn	Zr	Mg
Composition	2.58	0.54	0.54	Balance

**Table 2 materials-15-03556-t002:** Chemical composition of intermetallic phases in Figure 10 (in at.%).

Area	Element Content (at.%)				Area	Element Content (at.%)			
	Mg	Nd	Zn	Zr		Mg	Nd	Zn	Zr
A	99.7	0.3	0	0	M	99.36	0.28	0	0.37
B	97.93	1.43	0.64	0	N	98.42	1.58	0	0
C	99.46	0.54	0	0	O	96.12	3.88	0	0
D	97.51	1.96	0.53	0	P	98.78	0.95	0	0.27
E	99.62	0.38	0	0	Q	97.02	2.98	0	0
F	94.28	4.98	0.75	0	R	100	0	0	0
G	96.67	2.66	0.67	0	S	99.73	0.27	0	0
H	98.62	0.4	0.38	0.6	U	99.1	0.65	0.25	0
I	95.94	3.33	0.73	0	V	99.78	0.22	0	0
J	91.7	7.07	1.23	0	W	99.76	0.24	0	0
K	99.58	0.42	0	0	X	99.62	0.38	0	0
L	96.43	2.81	0.76	0	Y	97.56	1.86	0.59	0

**Table 3 materials-15-03556-t003:** The *D*_max_ and *D*_max_/*D*_av_ for different heat treatment states.

Time	470 °C		500 °C		530 °C	
	*D*_max_/μm	*D*_max_/*D*_av_	*D*_max_/μm	*D*_max_/*D*_av_	*D*_max_/μm	*D*_max_/*D*_av_
10 min	10.75	2.71	30.64	3.37	45.22	3.08
40 min	13.34	2.54	36.67	3.43	55.18	3.14
1 h	15.89	2.73	47.02	3.64	67.95	3.55
2 h	20.49	2.79	48.83	3.60	83.56	3.66
4 h	24.31	3.25	49.04	3.58	97.22	3.94
6 h	23.39	3.09	48.3	3.44	95.11	3.42
8 h	26.07	3.24	49.72	3.19	85.65	3.03
12 h	28.97	3.42	65.02	4.09	118.78	3.80
24 h	39.15	4.46	87.85	5.02	132.98	3.93

## Data Availability

The data presented in this study are available on request from the corresponding author. The data are not publicly available due to their association with an ongoing study.

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
