# Peer review of "The Microstructural Evolution and Grain Growth Kinetics of Fine-Grained Extruded Mg-Nd-Zn-Zr Alloy"

_materials, 2022, doi:10.3390/ma15103556_

Round 1

Reviewer 1 Report

Dear Authors, 

my comments on your work are as follows:

  1. The Introduction should be extended, including more references from last 2-3 years dealing with the topic. 
  2. Information about the apparatus, experimental and measuring (e.g. SEM, optical microscope), is missing. 
  3. How were the dimensions measured and histograms obtained?
  4. How many samples were tested? 
  5. Figure 5 contains the data from the calculations. But the formulas are given in next chapter. I think you should consider presenting the theoretical part before experiments. 
  6. The practical aspects of this research should be underlined. 
  7. "The average grain size calculated theoretically is closest to the actual 
    measured value only at 530°C, and the grain growth trend is basically the same" - it means that the results are close to the existing models. What is the novelty of your work?
  8. Further research should be indicated in the conclusions. 

Regards, 

Reviewer

Reviewer 2 Report

The subject of the paper is one of interest with well-documented observations and supported by the experimental program. The paper is well written and completely meets the standards of a scientific paper.

Author Response

Thank you for your comment

Reviewer 3 Report

Notes on the article of Xueyan Jiao, Liqiang Zhan, Gang Wang, Jin Ding and Jianlei Yang “The microstructural evolution and grain growth kinetics of fine-grained extruded Mg-Nd-Zn-Zr alloy”

The paper reports results of studying of the structure of the extruded alloy Mg-Nd-Zn-Zr after heating at different temperatures (from 470 °C to 530 °C). The authors report that the transition from normal to abnormal grain growth occurs earlier with increase in heating temperature. The presence of second phase inhibits the grain growth, but it causes the change of the normal type of grain growth to abnormal one. The authors did a painstaking research with a careful study of alloy structure at different temperatures.The results of this article have the high importance for research ofmagnesium alloys. This is an interesting and well-written report, which should be published after revisions that are listed below:

1) The authors claims "After preheating at 400°C for 9 h, a large number of Mg12Nd phases were still prуserved at the grain boundaries and the rod-like Zr-Zn phases were produced inside the grains [7, 15-16]". What about stoichiometry of the rod-like Zr-Zn phases?

2) The authors should explain in more detail by what signs they determined the transition point from normal to abnormal grain growth.

3) The designation of zones (A, B, C ...) in Figure 10 is difficult to read.

4) Typo should be corrected:

- P. 12, line 312. It should be written “Mg12Nd” instead of “Mg12Nd”.

Round 2

Reviewer 1 Report

Dear Authors, 

Please reply to my answers according to the standards of MDPI:   REVIEWER's COMMENT ...........   ANSWER: THE ADDED/MODIFIED TEXT (marked in red), and the explanation.   Regards,   Reviewer
